# Analysis of Healthy Lifestyle Habits and Oral Health in a Patient Sample at the Dental Hospital of the University of Barcelona

**DOI:** 10.3390/ijerph18147488

**Published:** 2021-07-14

**Authors:** Aina Torrejon-Moya, Beatriz Gonzalez-Navarro, Elisabet Roca-Millan, Albert Estrugo-Devesa, José López-López

**Affiliations:** 1Faculty of Medicine and Health Sciences (Dentistry), University of Barcelona, 08907 L’Hospitalet de Llobregat, Spain; aina.torrejon@gmail.com (A.T.-M.); erocamil@gmail.com (E.R.-M.); 2Oral Health and Masticatory System Group (Bellvitge Biomedical Research Institute), IDIBELL, Department of Odontostomatology, Faculty of Medicine and Health Sciences (Dentistry), University of Barcelona, 08907 Barcelona, Spain; beatrizgonzaleznavarro@gmail.com (B.G.-N.); albertestrugodevesa@gmail.com (A.E.-D.)

**Keywords:** oral health, lifestyle, healthy habits, weight, body mass index, periodontal disease, cardiovascular disease

## Abstract

The association between general health and oral health has been studied over recent years. In order to evaluate the lifestyle and the presence of healthy habits, a descriptive observational study was conducted from December 2018 to April 2019 with 230 patients, aged from 18 to 65 years old, that attended the Dental Hospital of the University of Barcelona for the first time. A total of 230 participants were considered, 98 (43%) were men and 132 (57%) were women, with a mean age of 37 years old. Our hypothesis was that patients with healthy habits had a better oral status in comparison with patients with bad lifestyle habits. No statistically significant results were found regarding oral hygiene between gender, smokers and patients with systemic pathology. Regarding a healthy lifestyle (High adherence to dietary intake), no statistically significant results were found. No significant differences were found regarding physical activity between male and female patients. Our hypothesis wasn’t confirmed; therefore, we cannot conclude that patients with healthy habits have better oral status in comparison with patients with bad lifestyle habits. Consequently, more prospective longitudinal studies should be carried out.

## 1. Introduction

The American Heart Association presented seven health parameters to reduce the risk of cardiovascular diseases: no smoking, being physically active, normal blood pressure, blood glucose, cholesterol levels, normal weight and a healthy diet [1].

In addition, obesity and other systematic diseases such as diabetes mellitus, hypertension and cardiovascular disease, increase across populations along with the degree of their development [2]; however, low socioeconomic status has been associated with a higher risk of both cardiovascular diseases and diabetes [3].

Several studies [2,3,4] agree that diet plays an important role in systemic disease prevention and life expectancy, along with physical activity, which should be established as a goal in order to maintain a healthy body.

Cardiovascular disease can be measured by indicators such as body mass index (BMI), waist circumference and waist-hip ratio [5,6]. As BMI increases, so do comorbid conditions. Consequently, physical activity and physical fitness are modifiers of mortality and morbidity related to obesity [3], which is a multifactorial disease that can be modulated by eating habits [4] and physical activity [3].

An association between general health and oral health has been studied during the recent years [7], in the same way previous studies showed an association between obesity and oral health [8,9].

The oral microbial composition is different between obese and nonobese patients, which could mean that there is a relation between oral bacteria and obesity, although glycemic control was not associated with oral bacteria [10].

This association occurs not only during adulthood, since obese children, for example, tend to have earlier teeth eruption [11].

Like obesity, clear differences in oral health have been shown in tobacco users. Among current smokers and never smokers, current smokers have poorer oral health and more oral problems than former smokers or never smokers [12].

The association between obesity and periodontitis is consistent, with an increased risk of periodontitis in overweight individuals. Although the pathophysiology remains unclear it is probably due to chronic inflammation and oxidative stress [13].

The prevalence and the severity of the periodontal disease are related to overweight and the treatment response occurs in all age groups [14].

The bidirectional association between periodontitis and diabetes is well known [15], because of oral dysbiosis and modified healing processes. Nevertheless, periodontitis is associated with microvascular and macrovascular complications [16]. Diabetic retinopathy, which is a major complication in diabetic patients, is higher in patients with type II diabetes and fewer teeth [17].

Heart disease has been associated with periodontal disease several times over the past years [18], with an increase of cardiovascular disease and strokes reported in patients with periodontitis [19].

Accordingly, this study was conducted to answer the following question: does lifestyle habits affect the oral health status of patients who attend the Dental Hospital of the University of Barcelona?

Subsequently, we set out to evaluate the lifestyle and presence of healthy habits of a group of patients who attended as their first visit to the Dental Hospital of the University of Barcelona.

We hypothesized that patients with healthy habits would have better oral status in comparison with patients with bad lifestyle habits.

Other specific objectives were established, such as determining the patient’s eating habits, establishing Body Mass Index (BMI), determining the level of physical activity, determining the patient’s oral hygiene status, as well as determining the number of caries, apical periodontitis and periodontal state of the patient through an orthopantomography.

## 2. Materials and Methods

### 2.1. Study Design 

A descriptive observational study was conducted from December 2018 to April 2019 with 230 patients aged from 18 to 65 years old that attended the Dental Hospital of the University of Barcelona for the first time. Written informed consent was obtained from the participants.

### 2.2. Research Ethics

The study protocol conforms to the ethical guidelines of the 1975 Declaration of Helsinki, and the present analysis was approved by the Ethics Committee of the Faculty of Dentistry, University of Barcelona (Comité d’Ética de l’Hospital Odontològic ceic.hospitalodontologic@ub.edu) on 29 September 2017, with the protocol number 31/207.

### 2.3. Study Participants

The patients were invited to participate in this study. Additionally, individuals with missing information on any of the covariates included in the multivariable regression models were also excluded.

The examinations included an anthropometric evaluation, physical activity, dietary questionnaire and oral examination.

### 2.4. Anthropometric Evaluation

A standardized questionnaire was completed and covered professional status, oral health habits, last dental visit, public or private visit, smoking and drinking habits, general pathology and medication intake.

Anthropometric evaluation included measurements of weight and height to calculate body mass index (BMI) and abdominal perimeter (Table 1).

### 2.5. Physical Activity Questionnaire

Patients were asked if they practiced physical activity, if they walked, and in the case of an affirmative responses how many hours they walked per week. The same questions were asked regarding aerobic physical activity. If they gave an affirmative response, how many hours per week of aerobic physical activity was reported.

### 2.6. Dietary Questionnaire

A modified healthy nutrition questionnaire was completed following two guidelines and modifying that used by the Spanish Arteriosclerosis Society [20,21] (Table 2).

If the patient had nine or more points, the patient was considered as having high adherence, meaning dietary intake was good.

### 2.7. Dental Examination

The dental examination was carried out by one trained dentist.

The total number of teeth were evaluated, as well as the presence of periapical lesions and their locations. The criteria were assessed by previously published criteria in the Orstravik periapical index (PAI) [22].

The endodontic burden [23], which consists of the sum of apical periodontitis, root canal treatment and the oral inflammatory burden [23] of each patient, was assessed. Oral inflammatory burden is divided into four categories and consists of the combination of evaluation of both the endodontic burden and periodontal disease.

The periodontitis severity index (PSI) [24] was assessed, taking into consideration that a PSI <1 is considered as no periodontal disease, between 1 and 1.9 is mild periodontal disease, between 2 and 2.9 is moderate periodontal disease and >2.9 is severe periodontal disease. Total dental index (TDI) [25] was estimated, which ranges from 0 to 10, the value increasing with severity. The orthopantomography index (OPGI) [25] was evaluated, which is the sum of chronic apical periodontitis, tertiary cavities, vertical deep pockets, radiolucent furcation area, and pericoronitis.

### 2.8. Statistical Analyses

Descriptive statistics were calculated for each of the studied features. A t-test was used for continuous variables, depending on distribution. For linear associations, the Pearson or Spearman correlation coefficient was used. An α = 0.05 level was considered statistically significant for all analyses.

## 3. Results

This section is divided by subheadings and provides a concise and precise description of the experimental results, their interpretation and the experimental conclusions that can be drawn.

### 3.1. Study Population and Characteristics of the Study Participants

A total of 230 participants were considered, 98 (43%) were men and 132 (57%) were women, with a mean age of 37.1 years old. The evaluated parameters are summarized in Table 3.

The mean BMI was 24.44. There were 124 (53.48%) patients with a normal BMI and 101 (43.91%) patients high a higher BMI.

#### 3.1.1. Systemic Pathology

Regarding systematic pathology, 10 (4.35%) patients reported being diabetic, 21 (9.13%) patients reported hypercholesterolemia, and 18 (7.83%) reported being hypertense. Patient medication was investigated. Four (1.77%) patients used antiaggregant medication, 9nine (3.91%) patients used hypocholesterolemia medication, nine (3.91%) used anxiolytic medication, 15 (6.52%) used a contraceptive pill, 21 (9.13%) used antihypertensive medication and 10 (4.35%) used antidepressant medication.

#### 3.1.2. Tobacco and Alcohol

Concerning the tobacco habit, 54 (23.48%) of the patients were current smokers, 29 (12.61%) were ex-smokers and 147 (63.91%) had never smoked. The mean number of cigarettes smoked per day was 10.4. Blond tobacco was more frequent (78–93.97%) and filters were used by most consumers (71–85.54%). The average number of years patients had been smoking was 9.78.

With regard to alcohol consumption, 16 (7.02%) patients were daily consumers, with a mean of 1.6 units of alcohol per day.

#### 3.1.3. Physical Activity

In relation to physical activity, 158 (68.70%) patients declared physical activity during their week. One hundred and seventy (74.24%) stated weekly walking with the mean of walking hours at 7.7 h per week, and 96 (41.74%) engaged in weekly aerobic activity such as soccer, tennis and gym activities, among others, with a mean of 4.31 h per week.

#### 3.1.4. Diet

A total of 230 questionnaires were evaluated, and 25 (10.87%) were classified as high adherence, meaning the participants had nine or more points in the questionnaire. Therefore, 205 (89.13%) participants had low adherence regarding the dietary intake questionnaire.

#### 3.1.5. Oral Examination

Regarding the oral evaluation, the results reported a mean of 26.04 teeth, 52 (22.71%). For periapical lesions, the PAI of each periapical lesion was calculated, and the mean was 3.0, following the Orstavik Index [22]. Mean PSI was 0.89, meaning no periodontal disease, and the TDI mean was 1.89. The OPGI mean was 1.77.

The mean of the endodontic burden was 0.8 and for OIB was 1.4.

#### 3.1.6. Correlation between Variables

No statistically significant results (*p* > 0.05) were found regarding oral hygiene, between gender, smokers and patients with systemic pathology. Regarding a healthy lifestyle (high adherence in dietary intake), no statistically significant results were found. No significant differences were found regarding physical activity between male and female patients.

Patients who had a healthier lifestyle reported less OPGI and TDI indexes (Table 4), although the results were not statistically significant. There were no statistically differences between apical lesions, PSI and cavities.

Patients who reported physical activity had less apical lesions (20.25–27.77%) and reported higher OPGI and TDI indexes (Table 5), although the differences weren’t statistically significant. No statistically differences were found regarding PSI and tertiary cavities.

## 4. Discussion

Answering our general objective; we cannot confirm that patients with wore lifestyle habits have worse oral health. In this study we analyzed several indicators of lifestyle habits such as, diet, physical activity, oral hygiene, smoking, obesity and cardiovascular parameters.

Regarding oral health, and in agreement with other studies [26], ours suggested that oral hygiene status was better in women than men, although the results weren’t conclusive.

Smoking is a habit related to many health disorders, and although some studies indicate higher percentages of smokers in men than in women [26,27], our study did not find statistically significant differences between genders. Furthermore, the association between smoking and an increased risk of periodontitis concluded in some studies [8,28,29,30] could not be confirmed.

This is probably because the mean age of our sample (37.1) indicated a very young population, whereas Holde et al. [30] established that the mean age of a sample with periodontitis was 47.3, and prevalence of periodontitis increased with age.

Holde et al. [30], also concluded that there was a 49.5% prevalence of periodontitis in a sample with a mean age of 47.3, in comparison our study, which showed 39.56%. Again, this difference can be explained by the mean age of our sample.

On the other hand, in our sample 23.48% of the patients were smokers, similar to the Spanish population (22.08%) [31].

Regarding obese and overweight patients, many cross-sectional studies [8,27] have indicated an association between obesity and periodontal disease [27] and having a larger number of teeth has been shown to be associated with a lower risk of obesity [27,28,29]. In our study this relation couldn’t be established, probably because the age of our sample was very young, and young patients tend to be more physically active [32]. On the other hand, and in agreement with our study, carious teeth showed no association with obesity [8,27].

According to the ENSE [31] 17.4% of adults are obese, and 37.1% are overweight. Our study showed similar results, with 43.91% of the patients being overweight and obese. It was suggested that obesity was more frequent in men than women. Another parameter related to obesity is BMI.

Taking this into consideration, Östberg et al. [27] found that dental visits and regular dental habits had an association with waist circumference and BMI. This fact should be taken into consideration since according to the ENSE [31] only 50.32% of the Spanish population visited a dentist during the past year. From all the evaluated population 18.18% visited the dentist during the last three months or less, 6.95% confirmed they had never been to the dentist, and women visited the dentist more frequently than men [31]. On the other hand, Shimazaki et al. [33] established that subjects with severe periodontitis had higher BMI; this is in agreement with other articles [34,35,36,37] that stated that weight-gain was directly associated with development of periodontitis.

Oral health has been associated with cardiovascular diseases since 1989 [25]. Since that time there has been several studies associating periodontitis and cardiovascular diseases [38,39,40].

Taking into account that physical activity is very much related to cardiovascular diseases, 35.28% of the Spanish population have a low level of physical activity during the week [31]. Our study suggested higher percentages of 55.1% in female patients and 44.94% in male patients; therefore, our results agree with the statement that the percentage is higher in woman versus men. Several studies [9,40,41,42] concluded that low physical activity is associated with periodontal disease. In our study this association couldn’t be confirmed because, as explained previously, the mean age in our study reflected a very young population, whereas periodontitis is associated with a much elderly population and increases with age [31]. Additionally, poor diet has been associated with periodontal disease [42]. We could not confirm this result for the same reason, i.e., periodontitis prevalence increases with age and our study involved a very young population.

The mean age of our study (37.1) indicated a very young population, providing probable bias in results. This made our results nonconclusive because younger people tend to have a healthier diet, do more physical activity, and their incidence of oral pathology is lower because it increases with age.

## 5. Conclusions

Our hypothesis wasn’t confirmed. Therefore we cannot conclude that patients with healthy habits have a better oral status in comparison with patients with bad lifestyle habits.

Consequently, more prospective longitudinal studies should be carried out for an optimal evaluation.

## Figures and Tables

**Table 1 ijerph-18-07488-t001:** Body Mass Index (BMI) and weight classification status. Adapted from Yang Q et al and Doll S et al. [1,2].

Weight Status	BMI (Kg/m^2^)
Underweight	<18.5
Normal range	18.5–24.9
Overweight	25–29.9
Obese	>30

**Table 2 ijerph-18-07488-t002:** Questionnaire provided to the patients. Adapted from Katsouyanni K et al. and Estruch R et al. [20,21].

Questions Provided	Yes	No	Point
Eat vegetables six or more days a week			
Eat fruits daily			
Eat whole foods (bread, cereals, rice ...)			
Eat legumes two or more days a week			
Eat nuts two or more days a week			
Eat olive oil daily			
Eat fish three or more times a week			
Eat less than four eggs a week			
Eat red meat or sausages less than three times a week			
Drink whole milk and dairy products			
Add salt to meals at the table			
Consume industrial pastries			
Eat industrial soft drinks (not light)			
Moderate alcohol abuse			

**Table 3 ijerph-18-07488-t003:** Summarized biometric parameters. BMI = Body Mass Index. Data presented as mean (SD) or *N* (%).

Biometric Parameters		Overall (*n* = 230) (%)	Healthy Lifestyle (High Adherence) (*n* = 25) (%)	*p*-Value	Self-Reported Physical Activity (*N* = 158) (%)	*p*-Value
Age		37.1 (15.51)	41.96 (16.36)	0.005	37.88 (16)	0.276
Gender (%)				0.077		0.274
	Male	98 (43)	8 (32)		71 (44.9)	
	Female	132 (57)	17 (68)		87 (55.06)	
BMI (%)		24.44 (9.76)	23.96 (5.58)	0.391	24.82 (4.37)	0.140
	Low BMI	5 (2.17)	0 (0)		2 (1.26)	
	Normal	124 (53.48)	18 (72)		89 (56.32)	
	High BMI	101 (43.91)	7 (28)		67 (42.4)	
Abdominal Perimeter		90.96 (24.56)	88.16 (12.19)	0.198	90.89 (25.32)	0.929
Current Smokers (%)		54 (23.48)	3 (12)	0.054	39 (24.68)	0.837
Hypercholesteremia		21 (9.13)	5 (20)	0.172	11 (6.96)	0.094
Statin medication		9 (3.91)	1 (4)	0.623	6 (3.79)	0.901

**Table 4 ijerph-18-07488-t004:** Oral data and healthy lifestyle.

Parameter Evaluated	Healthy Lifestyle (High Adherence) (*n* = 25) (%)	Unhealthy Lifestyle (Low Adherence) (*n* = 205) (%)	*p*-Value
OPGI	1.48	1.8	0.248
TDI	1.68	1.9	0.756
PSI	1	0.88	0.394
Apical lesions	6 (24)	46 (22.43)	0.826
Tertiary cavities	0.44	0.46	0.329

OPGI = Orthopantomography index. TDI = Total Dental Index. PSI = Periodontal Severity Index. Data presented as mean (SD) or *N* (%).

**Table 5 ijerph-18-07488-t005:** Oral data and weekly activity.

Parameter Evaluated	Weekly Physical Activity (*n* = 158) (%)	Lack of Physical Activity (*n* = 72) (%)	*p*-Value
OPGI	1.92	1.44	0.167
TDI	1.96	1.73	0.309
PSI	0.86	0.96	0.480
Apical lesions	32 (20.25)	20 (27.77)	0.262
Tertiary cavities	0.43	0.5	0.763

OPGI = Orthopantomography index. TDI = Total Dental Index. PSI = Periodontal Severity Index. Data presented as mean (SD) or *N* (%).

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
