# Peer review of "Analysis of Healthy Lifestyle Habits and Oral Health in a Patient Sample at the Dental Hospital of the University of Barcelona"

_ijerph, 2021, doi:10.3390/ijerph18147488_

Round 1

Reviewer 1 Report

In this observational study, the authors evaluated the lifestyle and presence of healthy habits in 230 subjects to see whether subjects with healthy habits have a better oral status, compared to subjects with bad lifestyle habits. The authors did not find any significant difference between the two groups.

There are several concerns with this manuscript.

The title of the manuscript talks about patient and the text also about patients in several places. What do these people suffer from? Also, is “Dental Hospital Barcelona University” the full name of the University?

When one reads the abstract, it is hard to understand what the authors are studying. The hypothesis is not clear. There is no mention of oral studies carried out or the results of the oral studies. This is main part of this study and it is disappointing that the authors have failed to mention it.

As the authors state often the mean age of the subjects used in this study reflects a very young population and cite this as a reason for the negative results obtained. Therefore, it is better to use a more representative subject population to study. Inclusion and exclusion criteria has not been provided. Did the authors exclude subjects with Sjogren’s syndrome from the study?

This work may be better suited for Journal of Negative Results.

Materials and methods need to have sub-headings.

The manuscript has been written poorly.

Some examples are provided below:

Based on the mentioned above?

the main question that conducted this study?

Percentages given as 49,5% some place while in other place given as 39’56%; 1’6 units?

that dental visits and regular habits showed an association with waist circumference and BMI- what habits?

only 50,32% of the population assisted to the dentist during last year?

Also, what population?

our mean age reflects a very young population?

The first sentence of the abstract has 67 words!

Author Response

Dear reviewer,

First of all, we would like to show our appreciation for your esteemed review in our paper.

Below, we answer all the questions that the reviewer indicated.

“The title of the manuscript talks about patient and the text also about patients in several places. What do these people suffer from?.”

# The patients may or may not suffer from a general disease but they all attended the Dental Hospital as patients.

“Also, is “Dental Hospital Barcelona University” the full name of the University?”

# Yes, the name is Dental Hospital – Barcelona University. But the title of the paper has been changed for a better understading.

“When one reads the abstract, it is hard to understand what the authors are studying. The hypothesis is not clear. There is no mention of oral studies carried out or the results of the oral studies. This is main part of this study and it is disappointing that the authors have failed to mention it.”

# The abstract has been re-written in order to give more importance to the results of the study.

“As the authors state often the mean age of the subjects used in this study reflects a very young population and cite this as a reason for the negative results obtained. Therefore, it is better to use a more representative subject population to study. Inclusion and exclusion criteria has not been provided. Did the authors exclude subjects with Sjogren’s syndrome from the study?”

# The only inclusion criteria were patients aged from 18 to 65 years old, that attended the Dental Hospital of the University of Barcelona for the first time. None of the 230 patients reported Sjogren’s syndrome, but it wasn’t an exclusion criteria.

“Materials and methods need to have sub-headings.”

# Sub-headings have been added to the materials and methods section

“The manuscript has been written poorly.

# The manuscript has been reviewed and all the mentioned examples have been modified.

Reviewer 2 Report

1. The results (especially regarding the association) are not explicitly presented in the abstract.

2. If IRB number or other ethical review number is available, please provide.

3. The method to assess the association between lifestyle and oral status needs to be elaborated so that we can assess the appropriateness of the method.

4. The title is misleading, because it only mentioned lifestyle and no "oral health."

5. How the results will be biased because of the sampling? (More specifically, hospital sample tend to be biased in many cases.) This should be at least discussed well in the discussion.

Author Response

Dear reviewer,

First of all, we would like to show our appreciation for your esteemed review in our paper.

Below, we answer all the questions that the reviewer indicated.

  1. The results (especially regarding the association) are not explicitly presented in the abstract.

The abstract has been modified to present the most important part of the results.

  1. If IRB number or other ethical review number is available, please provide.

There aren’t any more ethical numbers regarding our paper.

  1. The method to assess the association between lifestyle and oral status needs to be elaborated so that we can assess the appropriateness of the method.

The method is described in materials and methods. “. T-Test was used for continuous variables, depending on distribution. For lineal association the Pearson or Spearman correlation coefficient was used. An a = 0.05 level was considered statistically significant for all analyses.”

  1. The title is misleading, because it only mentioned lifestyle and no "oral health."

The title has been re-written to include the oral health concept.

  1. How the results will be biased because of the sampling? (More specifically, hospital sample tend to be biased in many cases.) This should be at least discussed well in the discussion.

This is now discussed in the discussion. “The mean age of our study (37,1) manifests a very young population, providing probably biased results and in all likelihood, this made our results non-conclusive. As younger people tend to have a healthier diet, have more physical activity and the incidence in oral pathology is lower, as it increases with age.”

Round 2

Reviewer 1 Report

This is a  much improved manuscript.

Only suggestion is the authors can remove (p > 0,05) from abstract and the text when saying "No statistically significant results (p > 0,05) were found"

Better to remove capitalization in the abstract for the word High  (High adherence in dietary intake)